# Critically Ill COVID-19 Patients Show Reduced Point of Care-Measured Butyrylcholinesterase Activity—A Prospective, Monocentric Observational Study

**DOI:** 10.3390/diagnostics12092150

**Published:** 2022-09-03

**Authors:** Florian Espeter, David Künne, Lena Garczarek, Henning Kuhlmann, Annabell Skarabis, Aleksandar R. Zivkovic, Thorsten Brenner, Karsten Schmidt

**Affiliations:** 1Department of Anesthesiology and Intensive Care Medicine, University Hospital Essen, University Duisburg-Essen, 45147 Essen, Germany; 2Department of Anesthesiology, Heidelberg University Hospital, 69120 Heidelberg, Germany

**Keywords:** COVID-19, sepsis, butyrylcholinesterase, point of care testing

## Abstract

A biomarker for risk stratification and disease severity assessment in SARS-CoV-2 infections has not yet been established. Point of care testing (POCT) of butyrylcholinesterase (BChE) enables early detection of systemic inflammatory responses and correlates with disease severity in sepsis and burns. In acute care or resource-limited settings, POCT facilitates rapid clinical decision making, a particularly beneficial aspect in the management of pandemic situations. In this prospective observational study, POCT-measured BChE activity was assessed in 52 critically ill COVID-19 patients within 24 h of ICU admission and on the third and seventh day after ICU admission. Forty (77%) of these patients required venovenous extracorporeal membrane oxygenation (vvECMO). In critically ill COVID-19 patients, BChE activity is significantly decreased compared with healthy subjects, but also compared with other inflammatory conditions such as sepsis, burns, or trauma. POCT BChE activity reflects the severity of organ dysfunction and allows prediction of 28-day mortality in critically ill COVID-19 patients. Implementing early POCT BChE measurement could facilitate risk stratification and support admission and transfer decisions in resource-limited settings.

## 1. Introduction

The clinical heterogeneity of symptom severity and unpredictable disease dynamics are major challenges for the medical treatment of Coronavirus disease 19 (COVID-19) patients. SARS-CoV-2 infection inflicts an excessive inflammatory response that underlies COVID-19 associated acute respiratory distress syndrome (ARDS) and multiple organ failure [1,2]. Biomarkers in COVID-19 patients enabling early risk stratification before clinical deterioration are still a matter of scientific debate. Diagnostic point-of-care testing (POCT) facilitates quick clinical decision making in acute care or resource-limited settings—a particularly advantageous aspect proposed for managing pandemic situations [3]. The cholinergic system plays an essential role in modulating an adequate immune response during systemic inflammation [4,5]. Serum cholinesterase (butyrylcholinesterase, BChE, also known as ChE or pseudocholinesterase), an acetylcholine hydrolyzing enzyme, has been repeatedly described as a clinically relevant and prognostic biomarker in acute inflammatory entities such as sepsis, trauma, and burns [6,7,8,9,10]. Therefore, BChE activity might reflect nonneuronal cholinergic activity during systemic inflammation [11,12].

Based on their observations in 26 COVID-19 patients in spring 2020, Nakajima et al. proposed conventionally-measured BChE levels to be a prognostic biomarker for severity and mortality in COVID-19 [13]. The role of cholinergic regulation in COVID-19 pathophysiology is not yet understood but has been proposed as a possible approach for future therapies [14,15,16]. Furthermore, Sridhar and Lakshmi have discussed the influence of different genetic expressions of BChE on the course of COVID-19 infection. It is possible that some variants lead to cholinergic dysfunction, and the prognosis of these patients could be negatively affected by an increased COVID-19-associated cytokine storm [17]. POCT-measured BChE activity can facilitate the early detection of systemic inflammation and correlates with disease severity and patient outcome in sepsis, trauma, and burns [7,9,10]. The integration of POCT-measured BChE in the emergency and critical care workflow could help in identifying patients at risk for deterioration and adverse outcomes [7,10]. Given the previous results of POCT-measured BChE in septic patients [9,18], we conducted a monocentric prospective observational study from April 2020 to April 2021 to evaluate an association between POCT-measured BChE and organ dysfunction severity in septic COVID-19 patients. Here, we report the results of POCT-measured BChE activity obtained from critically ill COVID-19 patients treated in a German tertiary medical center on a specialized acute respiratory distress syndrome (ARDS) intensive care unit (ICU).

## 2. Materials and Methods

This monocentric, prospective observational study was conducted from April 2020 to April 2021 at the ICU of the department of anesthesiology and intensive care medicine, University Hospital Essen, University Duisburg-Essen, Germany. The study was conducted in accordance with the Declaration of Helsinki, and the protocol was approved by the Ethics Committee of the medical faculty of the University Duisburg-Essen (ethic committee number: 20-9242-BO, date of approval: 20 April 2020). The study was registered in the German Clinical Trials Registry (German Clinical Trials Registry number: DRKS-ID: DRKS00022441).

### 2.1. Hospital Settings

The University Hospital Essen is a tertiary care medical center. The ICU operated by the department of anesthesiology and intensive care medicine is part of the West German Center for Infectious Diseases. Imbedded in the medical care strategy of University Hospital Essen, the department treated severely ill SARS-CoV-2 patients from the beginning of the COVID-19 pandemic. The department is a regional ARDS referral center with extracorporeal membrane oxygenation (ECMO) therapy specialization, with more than 100 ECMO procedures per year. The department offers a 24/7-critical care ARDS-transport system with a mobile ECMO unit (Cardiohelp ©, Getinge, Rastatt, Germany). The ECMO is capable of self-sufficient operation for several hours with a powerful battery and a mobile oxygen supply. The ECMO indication is made on site following patient evaluation and consultation, together with the treating physicians. After starting ECMO therapy, the patient is transferred to our center by intensive care transport via ground or air ambulance. These transports are performed by two physicians, one of whom is a board-certified senior physician in intensive care medicine, and two paramedics. For the intensive care unit, a senior physician with German specialist qualification in “intensive care medicine” is available on site during daytime and on call at night [19].

### 2.2. Patients Recruitement

Patients were consecutively included in this study from April 2020 until April 2021. Inclusion criteria consisted of SARS-CoV-2 infection verified with polymerase chain reaction (PCR) testing, sequential organ failure assessment score (SOFA score) > 2, and age over 18 years. Patients or their legal guardians gave informed consent to study participation. In cases in which patients without a legal guardian were unable to consent, an independent medical consultant approved study participation according to the regulations of the medical faculty of the University Duisburg-Essen. Following recovery, previously unable patients for consent were asked to consent in study participation. Exclusion criteria included patient refusal, probable discharge from the ICU within 72 h after admission, pregnancy, palliative therapy approach, and imminent death of the patient.

### 2.3. Patients Treatment

All patients admitted to the ICU were treated according to a standardized treatment protocol [19]. All patients received comprehensive laboratory diagnostics, microbiological examinations, invasive monitoring, and ultrasound examinations of the lungs and heart on admission [19]. The indication for endotracheal intubation and invasive ventilation was given by the attending intensivist [19]. Lung protective invasive ventilation was performed with a tidal volume of 6 mL/kg ideal body weight and optimal PEEP [19]. In COVID-19 patients with moderate or severe ARDS, prone positioning was performed with a duration of 16 h [19]. In case of clinical improvement (defined as improvement in PaO_2_/FiO_2_, improvement in dynamic lung compliance), prone positioning was continued for at least three days [18]. Pulmonary hypertension was assessed by echocardiography or pulmonary artery catheterization [19]. In cases with moderate/severe ARDS and pulmonary hypertension, we administered inhaled nitric oxide (NO) [19] and inhaled prostacyclin analogs. ECMO indication was based on Extracorporeal Life Support Organization (ELSO) recommendations and failure of conservative therapy and/or cardio-pulmonary deterioration [20]. The cannulation for ECMO therapy was performed bifemoral or femoro-jµgular.

Interdisciplinary COVID-19 treatment comprised the latest recommendations for COVID-19 therapy according to the current scientific level of knowledge, April 2020 until April 2021 [21]. Remdesivir, hydrocychloroquine, and reconvalescent plasma were used in our study population. All patients received heparin. For patients receiving vvECMO therapy, effective heparin anticoagulation was adapted accordingly.

### 2.4. Data Acquisition

Demographic data and clinical parameters were recorded as part of the clinical routine. The Charlson Comorbidity Index (CCI) was used to categorize preexisting conditions. Severity of illness was estimated using the modified SOFA score [22]. This modification allowed us to calculate the SOFA for patients in which the neurologic status was documented by the RASS rather than the Glasgow Coma Scale (GCS). For patients treated with venovenous extracorporeal membrane oxygenation (vvECMO), we adjusted the SOFA score and assigned an additional point. With this modification of the SOFA score, the theoretical maximum score increases from 24 to 25. ARDS was defined according to the Berlin definition of the American-European Consensus Conference [23]. For patients treated with venovenous extracorporeal membrane oxygenation (vvECMO), we adjusted the SOFA score and awarded one extra point. According to the recommendations reported by MacIntyre, all patients were followed for up to 28 days [24].

### 2.5. POCT BChE Measurements

Blood samples for the POCT BChE analysis were collected within 24 h following ICU admission (day 1) on day 3 and on day 7 following ICU admission, once per day. After routine blood gas analysis was performed, 10 uL of whole blood was collected from the sample for POCT-based measurement of BChE activity. Two independent POCT measurements were performed using the “LISA-CHE” POCT device (Version: TD 12.CHE 11.1–17.09; Dr. Franz Köhler Chemie GmbH, Germany; CE: according to the requirements of the directive 98/79 EG Declaration of Conformity according to Directive 98/79 EG Annex I/Annex III) according to the manufacturer manual. POCT BChE is expressed in kU/l.

#### BChE Data from Healthy Volunteers

Presented data from 40 healthy volunteers have previously been published by Zivkovic et al. [18].

### 2.6. Statistical Analysis

Data were entered into an electronic database (Microsoft Access, Microsoft Corp., Redmond, WA, WA) and evaluated using GraphPad Prism version 9.0 for Mac (GraphPad Software, La Jolla, CA, USA, http://www.graphpad.com, July 18, 2022). Data are presented as median with interquartile range (IQR). D’Agostino and Pearson’s omnibus normality test were used to check for Gaussian distribution. Statistical significance was tested using a Mann–Whitney test, and correlation analysis was performed using Spearman’s rank correlation test. A *p* value < 0.05 indicates statistical significance. The survival time analyzes were tested using Kaplan–Mayer curves and the logrank test (Mantel–Cox test).

## 3. Results

### 3.1. Characteristics of Critically Ill COVID-19 Patients

In total, 54 critically ill COVID-19 patients were included from April 2020 until April 2021. Two patients were subsequently excluded due to later refusal to consent to participate. According to the world health organization (WHO) COVID-19 severity classification, six patients were severely ill (oxygen requirement via mask or glasses) and 46 were critically ill (ventilation and/or vvECMO therapy) with COVID-19. Six patients were admitted directly to our ICU following triage in our emergency department or clinical deterioration on the COVID-19 intermediate care units of the University Hospital Essen. Forty-six patients were transferred to our department from referring hospitals due to severe COVID-19 with ARDS. VvECMO therapy due to severe ARDS was started by our mobile ECMO team in 25 external admissions. Demographic, clinical, and outcome patient data are depicted in Table 1.

### 3.2. BChE Activity Is Significantly Reduced in Critically Ill COVID-19 Patients

Compared to previously reported BChE activity levels in healthy volunteers (n = 40) [18], critically ill COVID-19 patients (n = 52) showed a significantly reduced BChE activity within 24 h following ICU admission (=day 1) (BChE activity is shown as median and interquartile range (IQR); healthy volunteers: 2.966 U/l (2.710–3.236) vs. critically ill COVID-19 patients: 1.311 U/l (941–1.629), *p* < 0.05; Figure 1A). Critically ill COVID-19 patients on vvECMO support on day 1 (n = 39) demonstrated an even more pronounced BChE activity reduction compared to healthy controls (healthy volunteers: 2.966 U/l (2.710–3.236) vs. critically ill COVID-19 patients on vvECMO support: 1.264 U/l (917–1.456), *p* < 0.05; Figure 1B). These findings were confirmed for respective comparisons on day 3 and day 7, revealing a sustained reduction in BChE activity in critically ill COVID-19 patients (day 3: 1.180 U/l (773–1.493) and day 7: 1.076 U/l (862–1.476); data not shown). We could not detect any significant difference in BChE activity between women and men in critically ill COVID-19 patients (women vs. men on day 1: 1341 U/I (815–1.530, n = 16) vs. 1305 U/I (981–1.697, n = 35), on day 3: 1210 U/l (735–1591, n = 16) vs. 1155 U/l (786–1.360, n = 36) and on day 7: 1063 U/l (839–1.391, n = 14) vs. 1076 U/l (856–1.496, n = 34)).

### 3.3. BChE Activity Correlates with Disease Severity in Critically Ill COVID-19 Patients

Analysis showed a negative correlation between BChE activity and SOFA scores at each observation time point following ICU admission (within the first 24 h: r = −0.60, on day 3: r = −0.35 and on day 7: r = −0.47; Spearman’s rank correlation test, Figure 2A–C).

### 3.4. Critically Ill COVID-19 Patients Requiring vvECMO Support Demonstrated a Lower BChE Activity Compared to Patients without vvECMO Support

Critically ill COVID-19 patients who did not require vvECMO therapy during the observation period of 28 days (n = 12) showed a significantly higher BChE activity compared to patients with vvECMO therapy on day 1 (critically ill COVID-19 patients without vvECMO support: 1.713 U/l (971–1.925) vs. critically ill COVID-19 patients on vvECMO support (n = 39): 1.264 U/l (916–1.456), *p* < 0.05; Figure 3A) and on day 3 (critically ill COVID-19 patients without vvECMO support: 1.447 U/l (1.125–1.754) vs. critically ill COVID-19 patients on vvECMO support (n = 40): 1.073 U/l (735–1.307), *p* < 0.05; Figure 3B). There was no difference in BChE activity between patients with and without vvECMO on day 7 after ICU admission (critically ill COVID-19 patients without vvECMO support (n = 7): 1.237 U/l (843–1.634) vs. critically ill COVID-19 patients on vvECMO support (n = 38): 1.020 U/l (833–1.323)). Thereafter we compared the BChE activity regarding the start time of the vvECMO therapy in critically ill COVID-19 patients during the observation period of 28 days. Patients admitted to our ICU on vvECMO support demonstrated a comparable BChE activity to those patients who required vvECMO therapy in the days following admission to our department (BChE activity in patients admitted with vvECMO (n = 25): 1.305 U/l (885–1.555) vs. patients that required vvECMO therapy in the days following admission (n = 15): 1.155 U/l (815–1.393); data not shown).

### 3.5. BChE Activity Reflects Level of Norepinephrine Support on Day Seven following ICU Admission

BChE activity corresponds to the level of norepinephrine support on day 7 following ICU admission. Cardiocirculatory compromised COVID-19 patients receiving high doses of norepinephrine (>0.3 µg/kg/min, n = 6) showed much lower BChE activity compared to those who needed low-dose norepinephrine support (<0.3 µg/kg/min, n = 42) (BChE activity in cardiocirculatory compromised COVID-19 patients with norepinephrine >0.3 µg/kg/min (n = 6): 530 U/l (373–1.275) vs. critically ill COVID-19 patients with norepinephrine <0.3 µg/kg/min (n = 42): 1.113 U/l (871–1.496), *p* < 0.05; Figure 4).

### 3.6. BChE Activity Correlates with C-Reactive Protein and Procalcitonin Levels

BChE activity correlated negatively with inflammation biomarkers C-reactive protein (CRP) (r = −0.54; Spearman’s rank correlation test; Figure 5A) and procalcitonin (PCT) (r= −0.67; Spearman’s rank correlation test; Figure 5B) on day 7 after ICU admission. We observed no correlation between BChE activity and CRP or PCT on day 1 or 3 and as well as white blood cell count (WBC) during the 7 days after ICU admission. Surprisingly, we could not find any correlation between BChE activity and Interleukin-6 (IL 6), an inflammation marker often associated with the COVID-19-related cytokine storm.

### 3.7. BChE Activity Measured within 24 h after Admission Identifies Survivors and Predicts Treatment Outcome

Critically ill COVID-19 patients with BChE activity greater than 1.557 U/l measured within the first 24 h after admission lived significantly longer than patients with BChE activity below 1.557 U/l (survival in days as median; BChE activity >1.557 U/l (n = 14): 28 days vs. BChE activity < 1.557 U/l (n = 38): 15 days, *p* < 0.05, Figure 6). 

## 4. Discussion

We present the first data on POCT-measured BChE activity in critically ill COVID-19 patients with ARDS and vvECMO support. Critically ill COVID-19 patients demonstrated a marked reduction in BChE activity compared to healthy volunteers. vvECMO support and vasopressor dependency were associated with an even more pronounced decrease in BChE activity. BChE levels correlated negatively with common inflammation biomarkers such as CRP and PCT. The key finding is that the observed low level of BChE activity reflects organ dysfunction severity in critically ill COVID-19 patients. In addition, higher BChE activity was associated with higher survival.

Serum cholinesterase or BChE has been repeatedly demonstrated to be a prognostic biomarker reflecting disease severity in acute inflammatory entities such as sepsis, trauma, and burns [7,9,10,18,25]. Previous studies showed a distinctive pattern in BChE activity change kinetics upon an inflammatory challenge, characterizing BChE as a negative acute phase protein. Following an initial decrease in BChE levels, sustained low levels were associated with higher morbidity and mortality, whereas an increase of serum BChE levels during the time-course of the disease was proposed to reflect recovery and better outcome [9,18]. During the SARS-CoV-2 pandemic in 2020, it became apparent that critically ill patients develop an excessive hyperinflammation that causes viral sepsis with ARDS [26]. Biomarkers for early detection of COVID-19 patients at risk are still the subject of scientific debate today [27,28]. Given the limited pharmacologic options and the short time window to attenuate hyperinflammation in SARS-CoV-2 infections, early recognition of patients at risk for deterioration is essential. Identifying COVID-19 patients at risk of developing a severe disease course remains a major challenge for attending physicians in intensive care units. Therefore, a simple bedside test for the inflammatory response could facilitate early clinical severity assessment and risk stratification. In resource-limited situations such as in a pandemic, the integration of POCT assays in the emergency and critical care workflow could be beneficial for risk stratification [3]. 

Based on their observation in 11 mild-to-moderate and 15 severe COVID-19 patients, Nakajima et al. already proposed in 2020 that early determination of cholinesterase could be used to assess COVID-19 prognosis at the time of hospital admission [13]. COVID-19 patients with severe symptoms had lower serum cholinesterase levels than patients with mild symptoms, and patients who later died showed lower serum cholinesterase levels than survivors [13]. Our group has previously demonstrated that POCT-based BChE measurements could facilitate early detection of an emerging inflammatory response in patients with sepsis, major trauma, and burns [7,10,18,25]. POCT-measured BChE levels correlated repeatedly with injury and organ dysfunction severity in critically ill patients and showed discriminative as well as prognostic power regarding morbidity and mortality [7,9,25]. The present results underline the benefit of a POCT-based BChE assessment of disease severity, even in COVID-19 patients. This is illustrated by the correlation between SOFA score and BChE activity in critically ill COVID-19 patients (Figure 2). Thus, POCT-based BChE measurement enables assessment of disease severity in critically ill COVID-19 patients quickly and with little effort immediately at the bedside.

We proposed that POCT-measured BChE activity complements conventionally available methods to identify patients at risk for adverse outcomes due to acute inflammatory diseases.

The first striking observations in this study are the low POCT BChE activity levels in our cohort of critically ill COVID-19 patients compared with our previous observations in septic, trauma, and burn patients as well as healthy subjects [7,9,10,18]. We have demonstrated that BChE activity is a predictor for 28- and 90-day mortality in patients with burns or sepsis. A BChE activity cut-off value of 1.661 kU/L at the clinical onset of sepsis with an ICU admission and BChE cut-off value of 2.644 kU/L in the emergency room following a severe burn trauma were predictive for a potentially fatal outcome [7,9]. To date, data on BChE levels as a biomarker in patients with ECMO therapy are limited. In 2014, Distelmaier et al. demonstrated a strong inverse association between BChE levels and short-term mortality as well as long-term mortality in patients undergoing vaECMO support after cardiovascular surgery [29]. 

BChE activity in this cohort of critically ill COVID-19 patients were consistently lower than the previously reported cut-off values in sepsis and burns. Nevertheless, POCT BChE measurements allowed reliable prognostic prediction of 28-day mortality in critically ill COVID-19 patients. Critically ill COVID-19 patients with BChE activity levels greater than 1.557 U/l measured 24 h after admission had a significantly higher 28-day survival rate (Figure 6). This finding highlights the prognostic power of POCT-based BChE measurements. Furthermore, BChE measurement provides a rapid and cost-effective prognostic information in critically ill COVID-19 patients within the first 24 h after ICU admission. Thus, the correlations of BChE activity with organ dysfunction, disease severity and survival are in line with previous results in sepsis, trauma, and burn patients [7,9,25]. Our observation that patients with vvECMO support showed an even greater reduction in BChE activity may further reflect the severity of organ failure. These results underline the ability of POCT BChE as a bedside biomarker for organ dysfunction severity. Furthermore, our study could demonstrate the predictive power of POCT BChE measurement in relation to mortality in critically ill COVID-19 patients. 

We observed a 64% 28-day mortality rate in the patient cohort of our study. Comparable mortality rates of up to 71 % in critically ill COVID-19 patients with vvECMO therapy were already observed in a large nation-wide German study with over 10,000 COVID-19 patients [30]. Accordingly, the analyses of an observational study at 26 German ECMO centers show that in high-volume centers such as ours (defined as specialized ECMO centers with more than 50 vvECMO therapies in 2019) 38% of COVID-19 vvECMO patients survived [31]. The timing of BChE measurement during SARS-CoV-2 infection in this study is a crucial factor for the interpretation of our data: previous studies evaluated BChE early in acute inflammatory entities, while most COVID-19 patients in our study may have been infected with SARS-CoV-2 for weeks and had already been treated in other intensive care units for several days before being transferred to our department. Therefore, the heterogeneity in our cohort of critically ill COVID-19 patients treated in a regional ARDS referral center regarding their individual illness time-course is a limitation of our study. Our results therefore present BChE activity levels in late COVID-19 following near fatal clinical deterioration with severe organ dysfunction.

BChE activity has been proposed as a possible readout of the cholinergic inflammatory response during inflammation [32,33]. Associations between cholinergic neuro-immunological function and COVID-19 hyperinflammation have been discussed recently [26]. Pomara and Imbimbo describe the association between older age and Alzheimer’s disease as well as a downregulated cholinergic anti-inflammatory pathway [34]. A deficiency or loss of cholinergic reaction could explain an increase in proinflammatory cytokines and the COVID-19-associated high mortality in elderly patients [34]. The authors do not specify at which age a patient has to be considered old, and therefore at what age a downregulated cholinergic anti-inflammatory pathway is to be assumed. In our study group, no patient had Alzheimer’s disease—the oldest study patient was 66 years old. We cannot provide data regarding the initial emergency or hospital admission BChE levels of our patients, but we presume that our results mirror the observations of Nakajima et al. in COVID-19 patients [13]. Furthermore, we propose that the low BChE levels and their trends might reflect a depleted immune response in our cohort of critically ill COVID-19 patients. Based on our results and previous data, such as those of Nakajima et al. [13], we sµggest that POCT BChE activity could facilitate early recognition of patients at risk for clinical deterioration, supporting hospital transfer decision making and the use of healthcare resources in patients suffering from COVID-19. 

Clinical scores for prognostic assessment, such as the SOFA score, are effective tools in critically ill ICU patients. However, they usually require a certain number of laboratory values (e.g., platelet count and cretainin value). These values are simple to determine but are associated with a certain time latency. In contrast, POCT-based BChE measurements might provide a reliable, cost-effective, and immediately available evaluation of disease severity as well as a prognostic risk stratification for COVID-19 patients.

Pharmacologic treatment approaches activating cholinergic anti-inflammatory mechanisms are discussed for COVID-19 pneumonia, and POCT BChE might represent a biomarker for the assessment of treatment efficiency [17]. In a major review, Hoover described in detail the possibilities of pharmacologically activating cholinergic anti-inflammatory mechanisms. Direct cholinergic stimulation with nicotine remained ruled out due to the unfavorable side effect profile [4]. Cholinergic immunomodulation with cholinesterase inhibitors such as physostigmine could potentially be used to treat severely ill COVID-19 patients [34]. An appropriate therapy response could then possibly be monitored by bedside BChE measurement.

The application of heparin in COVID-19 to prevent thromboembolic events is an integral part of current therapy. Paollisso et. al. discussed the anti-inflammatory effect of low molecular weight heparin [35]. An anti-inflammatory effect also appears to be present in patients with COVID-19, regardless of whether unfractionated or low molecular weight heparin is administered [36]. Patients in our study received unfractionated heparin; therefore, an anti-inflammatory effects of heparin that might influence BChE activity cannot be excluded.

We do not rule out further possible confounding effects on BChE activity in our patients. Hemodilution due to fluid resuscitation as well as an increased transcapillary loss, blood transfusions, impaired liver function (required for the BChE synthesis), and underlying genetic BChE phenotypes must be considered as limitations for data interpretation.

POCT-based BChE measurement has an as-yet unexploited potential to provide simple and cost-effective assessment of disease severity and prognosis in acute inflammatory disease entities and clinical settings. Further evaluation of this POCT concept in patients in emergency medicine and in different patient groups during critical care therapy, including pediatric as well as geriatrics, present future research opportunities. 

## 5. Conclusions

This study demonstrates POCT BChE activity as an inflammatory biomarker in critically ill COVID-19 ARDS patients. POCT BChE activity reflects the severity of organ dysfunction and enables a 28-day mortality prognosis of critically ill COVID-19 patients. Persistently low BChE activity levels might signal immune response exhaustion. Early POCT BChE measurement could facilitate risk stratification and help admission and transfer management decisions in resource-limited situations.

## Figures and Tables

**Figure 1 diagnostics-12-02150-f001:**
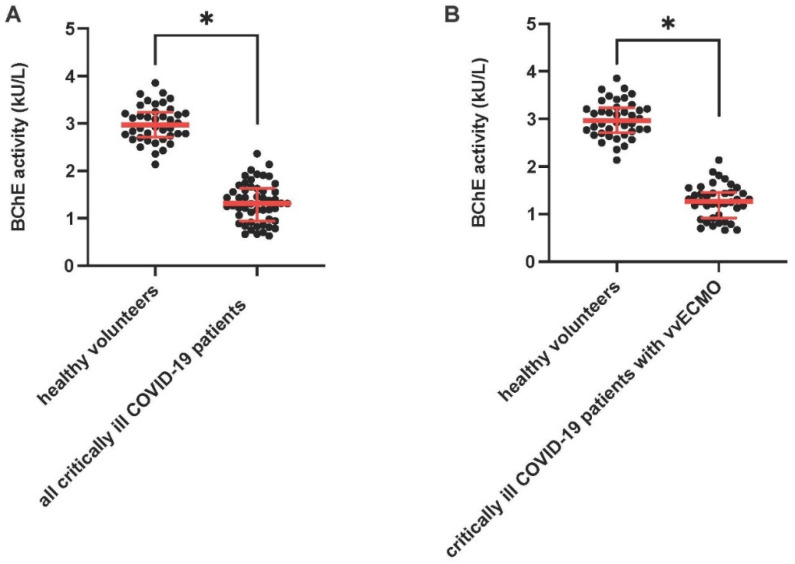
BChE activity in critically ill COVID-19 patients compared to healthy volunteers. Scatterplots represent POCT-based measurements of BChE activity in all critically ill COVID-19 patients (n = 52); (**A**) and in critically ill COVID-19 patients with vvECMO therapy (n = 39); (**B**) compared to previously published healthy volunteers (n = 40) [18] 24 h after ICU admission. * *p* < 0.05. Mann–Whitney U-Test.

**Figure 2 diagnostics-12-02150-f002:**
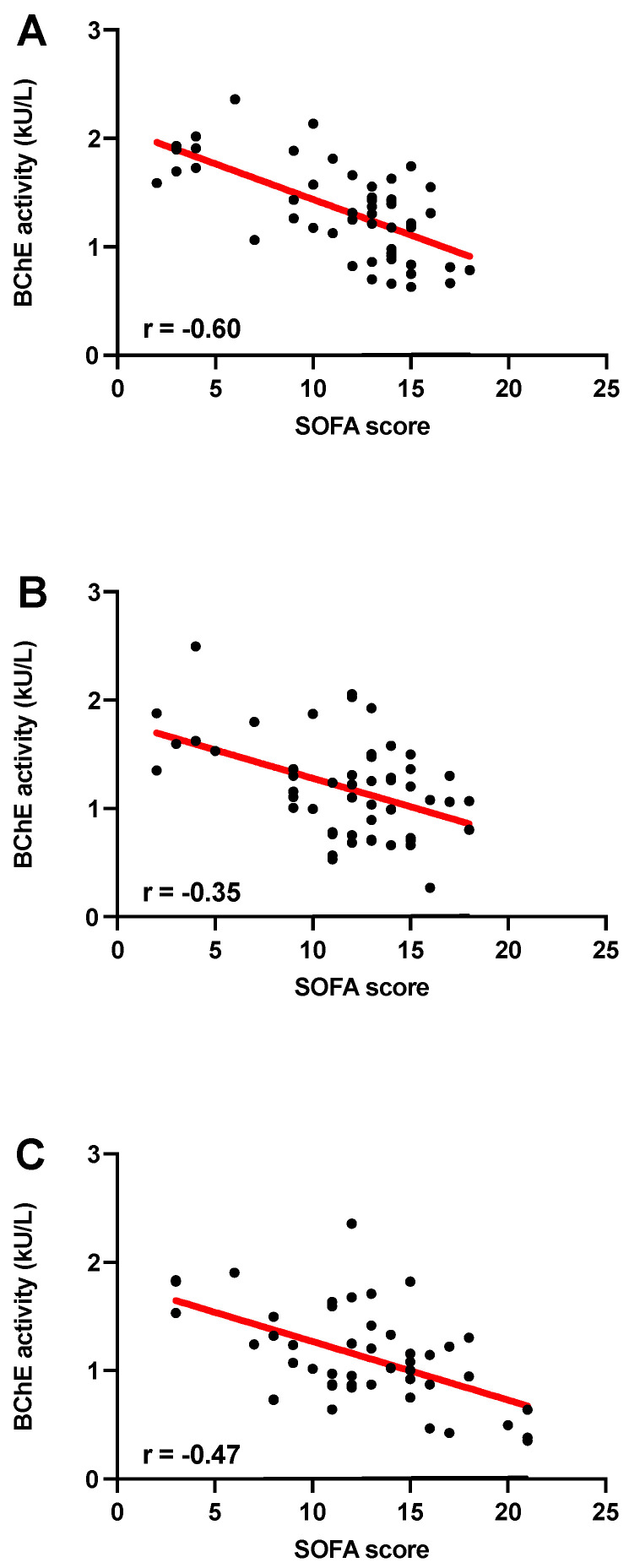
Correlation analysis of BChE activity with SOFA score-based assessment of disease severity in critically ill COVID-19 patients. Correlation of BChE activity with the SOFA score on day 1 (n = 52); (**A**); r = −0.60, on day 3 (n = 52); (**B**); r = −0.35 and on day 7 (n = 48); (**C**); r= −0.47; *p* < 0.05; Spearman’s rank correlation test.

**Figure 3 diagnostics-12-02150-f003:**
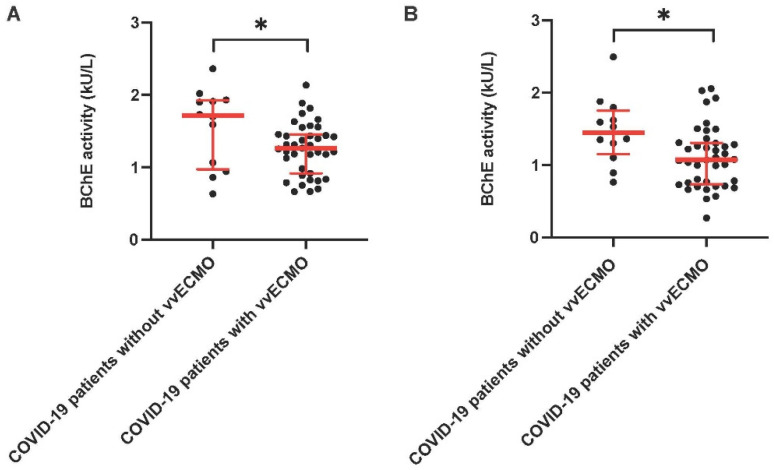
BChE activity in critically ill COVID-19 patients who did not require vvECMO therapy during the 28-day observation time compared to patients who required vvECMO therapy. Scatterplots represent the BChE activity in critically ill COVID-19 patients who did not require vvECMO therapy during the observation period of 28 days (n = 12) compared to patients with vvECMO on day 1 (1.713 U/L (971–1.925) vs. patients on vvECMO support (n = 39): 1.264 U/L (916–1.456)); (**A**) and on day 3 after ICU admission (1.447 U/L (1.125–1.754) vs. patients on vvECMO support (n = 40): 1.073 U/L (735–1.307)); (**B**).* *p* < 0.05 Mann–Whitney U-Test.

**Figure 4 diagnostics-12-02150-f004:**
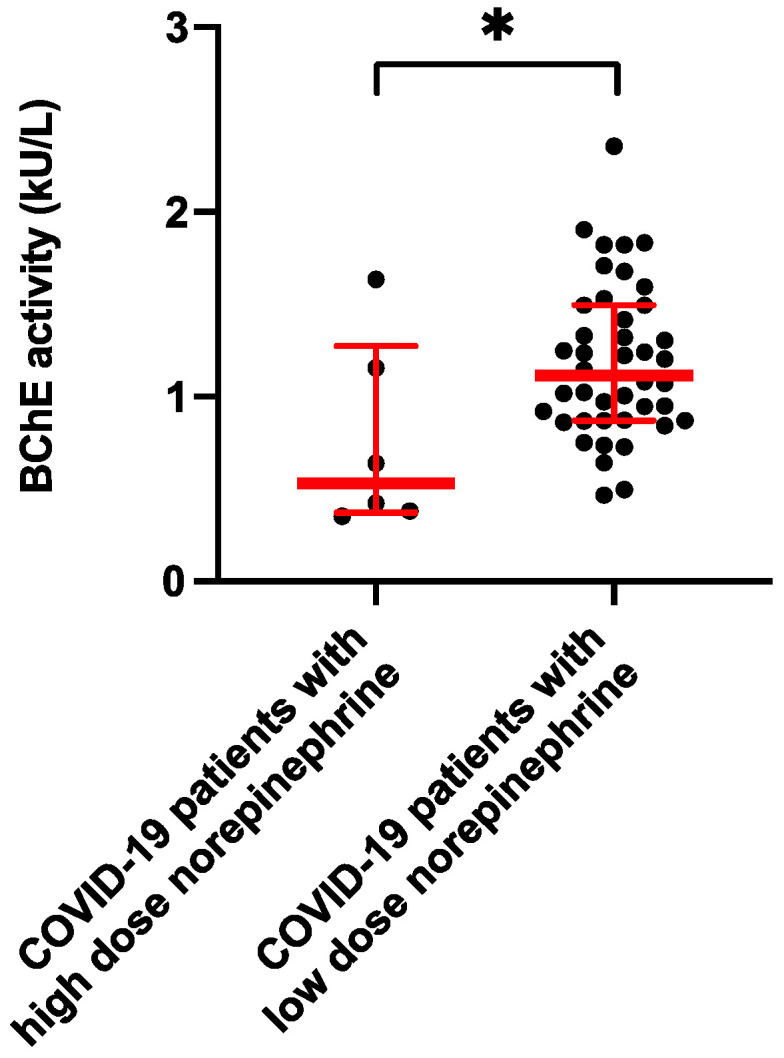
BChE activity reflects level of norepinephrine support on day 7 following ICU. Scatterplots represent the BChE activity in cardiocirculatory compromised COVID-19 patients with need for higher doses of norepinephrine (>0.3 µg/kg/min, n = 6) compared to critically ill COVID-19 patients with low-dose norepinephrine therapy (<0.3 µg/kg/min, n = 42) (BChE activity: 530 U/l (373–1.275) vs. 1.113 U/L (871–1.496)). * *p* < 0.05 Mann–Whitney U-Test).

**Figure 5 diagnostics-12-02150-f005:**
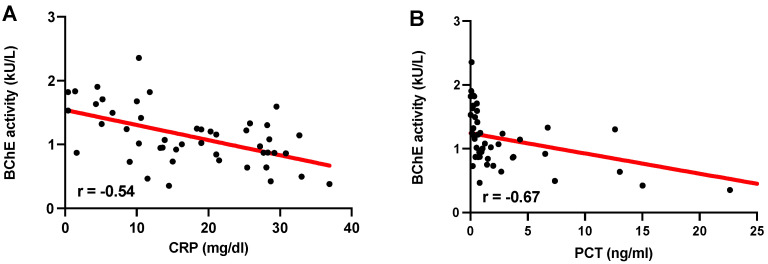
Correlation analysis of BChE activity with CRP and PCT levels in critically ill COVID-19 patients. Correlation of BChE activity with CRP (r = −0.54); (**A**) and with PCT (r = −0.67); (**B**) on day 7; *p* < 0.05, Spearman’s rank correlation test.

**Figure 6 diagnostics-12-02150-f006:**
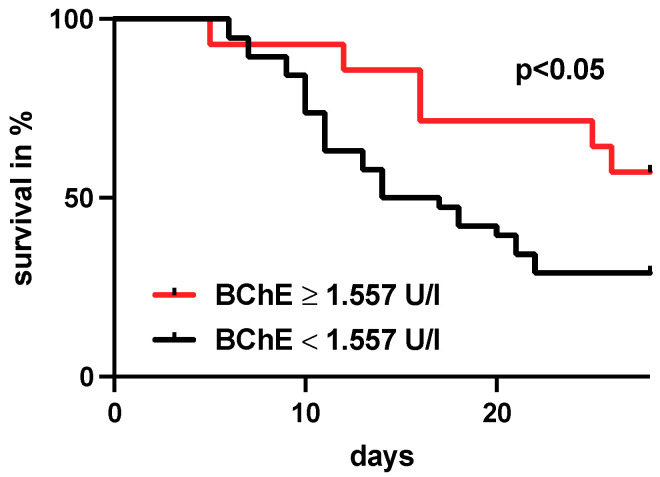
Kaplan–Mayer analysis showing the survival of critically ill COVID-19 patients with BChE activity > 1.557 U/l versus the survival of patients with BChE activity < 1.557 U/l. We demonstrated a survival advantage for BChE activity levels greater than 1.557 U/l measured 24 h after admission (BChE activity > 1.557 U/l in red (n = 14) vs. BChE activity < 1.557 U/l in black (n = 38), survival in days as median: 28 vs. 15, *p* < 0.05). The survival time analyzes were tested using Kaplan–Mayer curves and the logrank test (Mantel–Cox test).

**Table 1 diagnostics-12-02150-t001:** Characteristics of critically ill COVID-19 patients. * Median (interquartile range). CCI—Charlson Comorbidity Index; BMI—Body mass index; SOFA—the sequential organ failure assessment score; PCT—procalcitonin; IL 6—Interleukin 6; WBC—white blood cells; CRP—C-reactive protein; vvEVMO—venovenous extracorporeal membrane oxygenation; BChE—butyrylcholinesterase.

Characteristics of Critically ill COVID-19 Patients			
** *number of patients (n)* **			**52**
age (years)			59 (51–66)
male			36 (69%)
*clinical data*			
CCI			2 (2–3)
BMI (kg/m²)			31 (28–36)
at ICU admission			
spontaneous breathing oxygen dependent			6 (11.5%)
ventilated			46 (88.5 %)
non-invasive			6 (11.5%)
invasive			40 (77%)
vvECMO during ICU stay			40 (77%)
initiation at referring ICU			25 (48%)
initiation after ICU admission			15 (28%)
*outcome*			
28-day survivors			19 (36 %)
*disease severity*	*day 1*	*day 3*	*day 7*
SOFA Score *	13 (10–15) n = 52	12 (10–14) n = 52	12 (9–15) n = 48
*inflammation parameters*	*day 1*	*day 3*	*day 7*
PCT * (ng/mL)	1.3 (0.45–2.3)	0.7 (0.3–1.5)	0.8 (0.3–2.8)
WBC * (nL^−1^)	11.9 (9.3–17.0)	12.9 (9.0–15.5)	13.4 (9.9–21.0)
CRP * (mg/L)	22.2 (12.8–29.4)	13.9 (7.4–22.5)	15.9 (9.2–27.6)
IL 6 * (pg/mL)	154.5 (53.2–403.1)	140.0 (71.2–404.6)	243.3 (115.8–687.0)
BChE * (U/L)	1.308 (923–1.618)	1180 (773–1.493)	1076 (862–1.476)

## Data Availability

The datasets generated during and/or analyzed during the current study are available from the corresponding author on reasonable request.

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
