# Peer review of "Critically Ill COVID-19 Patients Show Reduced Point of Care-Measured Butyrylcholinesterase Activity—A Prospective, Monocentric Observational Study"

_diagnostics, 2022, doi:10.3390/diagnostics12092150_

Round 1

Reviewer 1 Report

This paper studies Critically ill COVID-19 patients show reduced point of care measured butyrylcholinesterase activity – a prospective, monocentric observational study. It could facilitate risk stratification and support admission and transfer decisions in resource-limited settings. This is an interesting paper that could be a potentially publishable subject. There are some general and main weaknesses that should be addressed in this paper. Therefore, I suggest the authors resubmit it after a Major revision. My suggestions are:

General comments:

1. Your introduction is too short. Meanwhile, please consider a literature review part in the introduction as well. Please separate the introduction and literature review.

2. The paper should be revised to include at least 10 recent high-quality COVID-19 references including:

- Influence of butyrylcholinesterase on the course of COVID-19

In vitro Antioxidant and Anticholinesterase Activities of Senecio massaicus Essential Oil and Its Molecular Docking Studies as a Potential Inhibitor of Covid-19 and Alzheimer’s Diseases

- Impairment of the cholinergic anti-inflammatory pathway in older subjects with severe COVID-19

The liver in times of COVID-19: What hepatologists should know

An integrated artificial intelligence model for efficiency assessment in pharmaceutical companies during the COVID-19 pandemic

COVID-19 and the liver

Pathophysiological mechanisms of liver injury in COVID-19

- COVID-19 and liver transplantation

-  A Novel Hybrid Parametric and Non-Parametric Optimisation Model for Average Technical Efficiency Assessment in Public Hospitals during and Post-COVID-19 Pandemic

Can use of hydroxychloroquine and azithromycin as a treatment of COVID-19 affect aquatic wildlife? A study conducted with neotropical tadpole

3. Your subsections for material and methods are too short. You have 6 subsections that are too short. Please explain more. In particular data acquisition and Patients Treatment.

4. The quality of English needs to be improved across the paper. Also, the scientific terms pertinent to your topic.

5. You have mentioned figure 6 twice in figure tilted please remove one of them and remove this kind of repetitive mistake in your manuscript.  

Main comments:

1. In lines 72-74you have mentioned:

"The department offers a 24/7-critical care ARDS-transport system with a mobile ECMO unit enabling initiation of ECMO at the referring hospital and transfer of the patient with or without ECMO via ground or air ambulance"

Please explain more about the critical care ARDS-transport system with a mobile ECMO. The critical care ARDS-transport system is so important and this part needs more clarification. In this part, only a short explanation and referring to one reference is not enough.

2. In lines 151-153 you have mentioned:

"We could not detect any difference in BChE activity between women and men in critically ill COVID-19 patients (data not shown)"

Why you could not detect any difference in BChE activity between women and men in critically ill COVID-19 patients? Please explain more about these results. 

3. SOFA scores in figure 2 and comparison among the suggested diagram need more discussion and clarification.

4. Kaplan Mayer's analysis in figure 6 needs more clarification and discussion.

5. In lines 245-333 you have provided other references in your discussion part. You should discuss your approach, limitations, and future studies in this part. It is true that you have compared other similar studies, but you need to transfer most of this part to the literature review section. 

Author Response

Response to Reviewer 1 Comments
This paper studies Critically ill COVID-19 patients show reduced point of care measured
butyrylcholinesterase activity – a prospective, monocentric observational study. It could facilitate risk
stratification and support admission and transfer decisions in resource-limited settings. This is an
interesting paper that could be a potentially publishable subject. There are some general and main
weaknesses that should be addressed in this paper. Therefore, I suggest the authors resubmit it after
a Major revision.
Thank you very much for critically reviewing our manuscript. We have revised our manuscript
accordingly. This is a point-to-point response to your comments. In the manuscript, revised parts are
presented in red font colour.

General comments
Point 1:
Your introduction is too short. Meanwhile, please consider a literature review part in the
introduction as well. Please separate the introduction and literature review.
Response 1: Thank you for this comment. We provide a focused introduction to the subject of our
work. For this purpose, we refer to important references of the past as well as current challenges.
Basically, our work is an original research manuscript. Our intention was not to write a literature
review. The structure of our manuscript follows the guidelines of Diagnostics (Abstract, Introduction,
Materials and Methods, Results, Discussion). For this reason, we would like to refrain from a separate
literature review within the introduction. Nevertheless, we have reviewed your literature
suggestions and are glad to add the work of Sridar and Laksmi to the introduction.

Point 2: The paper should be revised to include at least 10 recent high-quality COVID-19 references
including:
Response 2: We would like to thank you for your literature suggestions. We added “Influence of
butyrylcholinesterase on the course of COVID-19” by Sridar and Laksmi et al to our introduction and
discussion. In addition, “Impairment of the cholinergic anti-inflammatory pathway in older subjects
with severe COVID-19” by Pomara and Imbimbo was included in the context of possible new
approaches to COVID-19 therapy. It now reads as follows:
1. In ine 45-49:
Furthermore, Sridhar and Lakshmi discuss the influence of different genetic expressions
of BChE on the course of COVID-19 infection. It is possible that some variants lead to cholinergic
dysfunction and the prognosis of these patients could be negatively affected by an increased COVID-
19 associated cytokine storm.
2.
In line 358-365: Pomara and Imbimbo describe the association between older age and Alzheimer's
disease as well as a downregulated cholinergic anti-inflammatory pathway. A deficiency or loss of
cholinergic reaction could explain an increase in proinflammatory cytokines and the COVID-19
associated high mortality in elderly patients (Pomara and Imbimbo). The authors do not specify at
which age a patient has to be considered old and herefore, at what age a downregulated cholinergic

anti-inflammatory pathway is to be assumed. In our study group, no patient had Alzheimer's disease
- the oldest study patient was 66 years old.

Point 3: Your subsections for material and methods are too short. You have 6 subsections that are too
short. Please explain more. In particular data acquisition and Patients Treatment.
Response 3: Thank you for this comment. We have adapted the sections data acquisition and patient
treatment. We now describe in detail the treatment of COVID-19 patients. Furthermore, in the data
acquisition section we now explain the adaptation of the SOFA score to our conditions. It now reads
as follows:
1. In line 98-112: All patients received comprehensive laboratory diagnostics, microbiological
examinations, invasive monitoring, and ultrasound examinations of the lungs and heart on admission
[18]. The indication for endotracheal intubation and invasive ventilation was given by the attending
intensivist [18]. Lungprotective invasive ventilation was performed with a tidal volume of 6 ml/kg
ideal body weight and optimal PEEP [18]. In COVID-19 patients with moderate or severe ARDS,
prone positioning was performed with a duration of 16 hours [18]. In case of clinical improvement
(defined as improvement in PaO2/FiO2, improvement in dynamic lung compliance), prone
positioning was continued for at least three days [18]. Pulmonary hypertension was assessed by
echocardiography or pulmonary artery catheterization [18]. In cases with moderate/severe ARDS and
pulmonary hypertension, we administered inhaled nitric oxide (NO) [18] and inhaled prostacyclin
analogs. ECMO indication was based on Extracorporeal Life Support Organization (ELSO)
recommendations and failure of conservative therapy and/or cardio-pulmonary deterioration [19]. The
cannulation for ECMO therapy was performed bifemoral or femoro-jugular.
2.
In Line 121-126: This modification allowed us to calculate the SOFA for patients in which the
neurologic status was documented by the RASS rather than the Glasgow Coma Scale (GCS). For
patients treated with venovenous extracorporeal membrane oxygenation (vvECMO), we adjusted the
SOFA score and assigned an additional point. With this modification of the SOFA score, the theoretical
maximum score increases from 24 to 25.

Point 4: The quality of English needs to be improved across the paper. Also, the scientific terms
pertinent to your topic.
Response 4: Thank you for this comment. The manuscript and the current draft have been read and
corrected by a native spekaer.

Point 5: You have mentioned figure 6 twice in figure tilted please remove one of them and remove
this kind of repetitive mistake in your manuscript.
Response 5:
Thank you for this comment. We have removed the duplicate label from Figure 6. We
rechecked the entire manuscript for incorrect or duplicate labeling.

Main comments
Point 1:
In lines 72-74you have mentioned: "The department offers a 24/7-critical care ARDS-transport
system with a mobile ECMO unit enabling initiation of ECMO at the referring hospital and transfer
of the patient with or without ECMO via ground or air ambulance"
Please explain more about the critical care ARDS-transport system with a mobile ECMO. The critical
care ARDS-transport system is so important and this part needs more clarification. In this part, only
a short explanation and referring to one reference is not enough.

Response 1: Thank you for this comment. We agree with your opinion that a specific COVID-19
ARDS treatment/ARDS-transport system is a very relevant topic. Nevertheless, we do not want to
overwhelm the reader with too much information on this topic as it is not the main focus of our study.
Keeping this in mind, we have adapted the description of treatment as well as the work of the 24/7
ECMO team. It now reads as follows: Line 75-84:
The department offers a 24/7-critical care ARDStransport system with a mobile ECMO unit (Cardiohelp ©, Getinge, Rastatt, Germany). The ECMO is capable of self-sufficient operation for several hours with a powerful battery and a mobile oxygen supply. The ECMO indication is made on site following patient evaluation and consultation together with the treating physicians.
After starting ECMO therapy, the patient is transferred to our center by intensive care transport via ground
or air ambulance. These transports are performed by two physicians, one of whom is a board certified senior
physician in intensive care medicine, and two paramedics. For the intensive care unit, a senior physician with
german specialist qualification in "intensive care medicine" is available on site during daytime and on call at
night [18].

Point 2: In lines 151-153 you have mentioned:"We could not detect any difference in BChE activity
between women and men in critically ill COVID-19 patients (data not shown)"
Why you could not detect any difference in BChE activity between women and men in critically ill
COVID-19 patients? Please explain more about these results.
Response 2: Thank you for this comment. We now show the results of BChE activity from our female
and male COVID-19 patients at day 1, day 3, and day 7 (line 183-186). In principle, there seems to be
a difference in BChE activity between women and men in a healthy population (
Activity of
cholinesterases in a young and healthy middle-European population: Relevance for toxicology, pharmacology
and clinical praxis,
Karasova, 2017). However, we did not observe any differences in our severely ill
COVID-19 patients, so that we cannot prove a sex-specific effect. It is possible that BChE activity
converges during the course of severe inflammatory disease like COVID-19, so that no differences
are measurable.

Point 3: SOFA scores in figure 2 and comparison among the suggested diagram need more discussion
and clarification.
Response 3: Thank you for this comment. We now discuss the results of Figure 2 (SOFA score and
BChE activity) in more detail. It now reads as follows: (Line 306-310)
The present results underline the
benefit of a POCT based BChE assessment of disease severity even in COVID-19 patients. This is illustrated
by the correlation between SOFA score and BChE activity in critically ill COVID-19 patients (Figure 2). Thus,
POCT based BChE measurement enables assessment of disease severity in critically ill COVID-19 patients
quickly and with little effort immediately at the bedside.

Point 4: Kaplan Mayer's analysis in figure 6 needs more clarification and discussion.
Response 4: Thank you for this comment. We now discuss the results of Figure 6 (Kaplan Mayer's
analysis) in more detail. It now reads as follows: (Line 330-333)
This finding highlights the prognostic
power of POCT based BChE measurements. Bedside BChE measurement provides a rapid and cost-effective
prognostic information in critically ill COVID-19 patients within the first 24 hours after ICU admission.

Point 5: In lines 245-333 you have provided other references in your discussion part. You should
discuss your approach, limitations, and future studies in this part. It is true that you have compared
other similar studies, but you need to transfer most of this part to the literature review section.
Response 5: Thank you for this comment. We actually thought that we have sufficiently discussed
our results and the study approach. Nevertheless, according to your suggestion, we have extended
our discussion in some parts. In addition, we highlight possible future scientific questions regarding
POCT based BChE measurements. We kindly point out that we refrain from a detailed literature
review section - on the one hand due to the requirements of the journal (see comment above) and on
the other hand due to the positive feedback of the other two reviewers. It now reads as follows:
1. Line 373-378
: Clinical scores for prognostic assessment, such as the SOFA score are effective tools in
critically ill ICU patients. However, they usually require a certain number of laboratory values (e.g.
platelet count and cretainin value). These values are simple to determine but are associated with a
certain time latency. In contrast, POCT based BChE measurements might provide a reliable, costeffective and immediately available evaluation of disease severity as well as a prognostic risk
stratification for COVID-19 patients.
2.
Line 399-403: POCT based BChE measurement has an as yet unexploited potential to provide simple
and cost-effective assessment of disease severity and prognosis in acute inflammatory disease entities
and clinical settings. Further evaluation of this POCT concept in patients in the emergency medicine
and in different patient groups during critical care therapy including pediatric as well as geriatrics
present future research opportunities.

Reviewer 2 Report

In the submitted study, the authors carried out research on the impact of early detection of a biomarker-butyrylcholinesterase (BChE) as Point of care testing (POCT). They observed that in critically ill COVID-19 patients, BChE activity is significantly decreased compared with healthy subjects. They claim with supporting results that POCT BChE activity reflects the severity of organ dysfunction and allows the prediction of 28-day mortality in critically ill COVID-19 patients. Authors are also claiming that biomarker-butyrylcholinesterase (BChE) can be a great tool in risk stratification and support admission and transfer decisions in resource-limited settings. The study suits the requirement of society and could be helpful in still going pandemic. There are a few minor concerns that should be addressed by the authors.

  1. Section 2.5, butyrylcholinesterase (BChE) analysis was done directly on the blood or plasma was separated, and then analysis was carried out. Kindly clarify.
  2. The X-axis in figures 1, 3, and 4 should be written appropriately.
  3. Ref 20 looks incomplete.
  4. Do authors consider any impact of gender or ethnicity on their study? 

Reviewer 3 Report

Espeter F and coworkers evaluated the correlation between early detection of systemic inflammatory responses with disease severity in sepsis and burns. In this prospective observational study, point of care testing (POCT) measure and butyrylcholinesterase (BChE) activity were assessed in 52 critically ill COVID-19 patients within 24 hours of on the third and seventh day after ICU admission. The authors concluded that POCT and BChE activity reflect the severity of organ dysfunction and allow the prediction of 28-day mortality in critically ill COVID-19 patients. 

This is a prospective study with a small number of patients admitted in ICU, consequently very high-risk. I have a few issues which deserve clarifications:

  1. Did the authors perform ECG? If yes can give the results of ECG because inflammation can induce arrhythmias and ECG alterations.
  2. What is the medical treatment of the patients? Did they treat with antiviral drugs, heparin, ……..?
  3. The authors should discuss the effects of the therapy on inflammation markers (please cite PMID: 32848743; PMCID: PMC7424043.
  4. The authors should also discuss the role of ECG in these patients (please cite: PMID: 33512742; PMCID: PMC7994985)

Author Response

 1
Response to Reviewer 3 Comments
Espeter F and coworkers evaluated the correlation between early detection of systemic inflammatory
responses with disease severity in sepsis and burns. In this prospective observational study, point of
care testing (POCT) measure and butyrylcholinesterase (BChE) activity were assessed in 52 critically
ill COVID-19 patients within 24 hours of on the third and seventh day after ICU admission. The
authors concluded that POCT and BChE activity reflect the severity of organ dysfunction and allow
the prediction of 28-day mortality in critically ill COVID-19 patients.
This is a prospective study with a small number of patients admitted in ICU, consequently very highrisk. I have a few issues which deserve clarifications.

Thank you very much for critically reviewing our manuscript. We have revised our manuscript
accordingly. Please find enclosed a detailed point-to-point response to your questions and comments.
In the manuscript, revised parts are presented in red font colour.

Point 1: Did the authors perform ECG? If yes can give the results of ECG because inflammation can
induce arrhythmias and ECG alterations.
Response 1: Thank you for this comment. We agree with you, that COVID-19 associated arrythmia
and ECG alterations are an interesting issue to explore. All critically ill COVID-19 patients received
permanent ECG monitoring. However, we did not explicitly collect this data. Therefore, we regret
that we are unable to offer results of ECG.

Point 2: What is the medical treatment of the patients? Did they treat with antiviral drugs, heparin,
……..?
Response 2: Thank you for this comment. COVID-19 specific therapy was interdisciplinary agreed
upon and included the latest recommendations according to current scientific knowledge.
Accordingly, not all patients received the same treatment. Remdesivir, hydroxychloroquine, and
reconvalescent plasma were used in our study population. In addition, all patients received heparin.
For patients with vvECMO therapy, we increased the heparin dosage accordingly. The related text
section now reads as follows (line 113-117):
Interdisciplinary COVID-19 treatment comprised the latest
recommendations for COVID-19 therapy according to the current scientific level of knowledge from April 2020
until April 2021 [20]. Remdesivir, hydrocychloroquine and reconvalescent plasma were used in our study
population. All patients received heparin. For patients receiving vvECMO therapy effective heparin
anticoagulation was adapted accordingly.

Point 3: The authors should discuss the effects of the therapy on inflammation markers (please cite
PMID: 32848743; PMCID: PMC7424043.
Response 3: Thank you for this comment. The application of heparin in COVID-19 to prevent
thromboembolic events is an integral part of current therapy. Although the patients in our study
received unfractionated heparin, a possible effect of heparin on the anti-inflammatory response of
BChE cannot be excluded. We now included this aspect in the discussion. It now reads as follows
(line 388-394):
The application of heparin in COVID-19 to prevent thromboembolic events is an integral part
of current therapy. Paollisso et. al. discuss an anti-inflammatory effect of low molecular weight heparin
(Paollisso et). An anti-inflammatory effect also appears to be present in patients with COVID-19, regardless of
whether unfractionated or low molecular weight heparin is administered (Tritschler et al.). Patients in our
study received unfractionated heparin, therefore anti-inflammatory effects of heparin that might influence
BChE activity cannot be excluded.

Point 4: The authors should also discuss the role of ECG in these patients (please cite: PMID:
33512742; PMCID: PMC7994985)
Response 4: Thank you for the literature suggestion and the idea to discuss COVID-19 associated
ECG alterations. As mentioned above, we do not have ECG data of our patients and therefore refrain
from discussing this issue.

Round 2

Reviewer 1 Report

The authors just answer all my comments except one main point. Please explain the structure of your paper at the end of the introduction. 

Author Response

Response to Reviewer 1 Comments,
Round 2
Point 1: The authors just answer all my comments except one main point. Please explain the
structure of your paper at the end of the introduction.
Response 1: Thank you very much for your pleasant review. The article's outline is based on the
requirements of the journal and corresponds to the generally accepted structure of a medical
manuscript. Therefore, we do not consider it necessary to present the outline at the end of the
introduction.

Reviewer 3 Report

My previous comments were not taken into account

Author Response

Point 1: My previous comments were not taken into account
Response 1: Unfortunately, we do not understand your evaluation of our revised manuscript. We
have answered your comments conscientiously and in detail. Accordingly, each of your
questions/comments were discussed point-by-point. We have included these point-by-point
responses to your comments again below as a precaution. We would appreciate your consideration
of the answers.